# Cause-Specific Stillbirth and Neonatal Death According to Prepregnancy Obesity and Early Gestational Weight Gain: A Study in the Danish National Birth Cohort

**DOI:** 10.3390/nu13051676

**Published:** 2021-05-15

**Authors:** Ellen Aagaard Nohr, Sanne Wolff, Helene Kirkegaard, Chunsen Wu, Anne-Marie Nybo Andersen, Jørn Olsen, Bodil Hammer Bech

**Affiliations:** 1Research Unit for Gynaecology and Obstetrics, Department of Clinical Research, University of Southern Denmark, 5000 Odense, Denmark; cwu@health.sdu.dk; 2Department of Oncology, Aarhus University Hospital, 8200 Aarhus, Denmark; sannelausenwolff@hotmail.com; 3Steno Diabetes Center Aarhus, Aarhus University Hospital, 8200 Aarhus, Denmark; helene.kirkegaard@rm.dk; 4Department of Public Health, University of Copenhagen, 1014 Copenhagen, Denmark; amny@sund.ku.dk; 5Department of Clinical Epidemiology, Aarhus University Hospital, 8200 Aarhus, Denmark; jo@clin.au.dk; 6Department of Public Health, Research Unit for Epidemiology, Aarhus University, 8000 Aarhus, Denmark; bhb@ph.au.dk

**Keywords:** fetal death, stillbirth, neonatal death, pregnancy, obesity, body mass index, gestational weight gain, gestational weight loss

## Abstract

Maternal obesity is associated with impaired fetal and neonatal survival, but underlying mechanisms are poorly understood. We examined how prepregnancy BMI and early gestational weight gain (GWG) were associated with cause-specific stillbirth and neonatal death. In 85,822 pregnancies in the Danish National Birth Cohort (1996–2002), we identified causes of death from medical records for 272 late stillbirths and 228 neonatal deaths. Prepregnancy BMI and early GWG derived from an early pregnancy interview and Cox regression were used to estimate associations with stillbirth or neonatal death as a combined outcome and nine specific cause-of-death categories. Compared to women with normal weight, risk of stillbirth or neonatal death was increased by 66% with overweight and 78% with obesity. Especially deaths due to placental dysfunction, umbilical cord complications, intrapartum events, and infections were increased in women with obesity. More stillbirths and neonatal deaths were observed in women with BMI < 25 and low GWG. Additionally, unexplained intrauterine death was increased with low GWG, while more early stillbirths were seen with both low and high GWG. In conclusion, causes of death that relate to vascular and metabolic disturbances were increased in women with obesity. Low early GWG in women of normal weight deserves more clinical attention.

## 1. Introduction

Among adverse outcomes related to maternal obesity, the increased risk of losing a child, whether it is in late pregnancy or shortly after birth, is probably the most devastating and tragic event. Today, a plenitude of studies contribute to the evidence on impaired survival in the offspring of mothers with obesity, both in late pregnancy and during the first month of life [1], but the underlying mechanisms are still poorly understood. While it is well-established that especially post-term pregnancies in women with obesity are at higher risk of antepartum intrauterine death [2,3], the evidence about other causes or subtypes of stillbirth or neonatal death associated with maternal obesity is less consistent. In a subpopulation of the Danish National Birth Cohort, we observed maternal overweight and obesity to be associated with antepartum stillbirth with placental dysfunction [3], which has later been supported by data from the USA [4]. In North American data from the 1980s, the highest excess mortality in infants of mothers with obesity was related to short gestation and low birth weight [5]. In contemporary data from Sweden, specifically neonatal deaths from birth asphyxia and other neonatal morbidities were associated with maternal obesity [6].

Animal studies suggest that a high fat diet may increase risk of placental dysfunction and stillbirth [7,8], which corresponds well with the increased focus on dietary interventions in pregnancy in women with obesity. However, large studies in humans of the association between total gestational weight gain (GWG) and stillbirth or neonatal death rather point to inadequate gain as a risk factor [9,10,11]. As total GWG may also reflect abnormal physiology leading to stillbirth or neonatal death rather than being a marker of nutrition [12], focusing on GWG in early pregnancy, which is known to be a period of building fat reserves [13], may be a better choice.

It is well accepted that causes for stillbirth and neonatal death overlap, and depending on clinical practice, a perinatal death may end up as a stillbirth or neonatal death. Therefore, it makes good sense to combine these two outcomes when investigating causes [14,15]. In the Danish National Birth Cohort, we used a combined classification system to categorize causes of all late stillbirths and neonatal deaths, based on a classification system suggested by Andersen et al. [16]. The aim of this study was to examine how the mother’s prepregnancy weight and early gestational weight gain were associated with cause-specific stillbirth and neonatal death.

## 2. Materials and Methods

We conducted a prospective cohort study based on data from the Danish National Birth Cohort (DNBC) [17]. Enrolment in the DNBC took place between 1996 and 2002 and participants were recruited at their first antenatal visit to the general practitioner, where pregnant women were invited to participate in four computer-assisted telephone interviews (two during pregnancy and two after delivery). Approximately 60% of invited women chose to participate in the DNBC. Inclusion was restricted to women who intended to carry the pregnancy to term and spoke Danish well enough to be able to complete oral interviews. In total, 100,413 pregnancies of 91,381 women were included in the cohort.

### 2.1. Stillbirth and Neonatal Death

Information regarding pregnancy outcomes was obtained from The Medical Birth Register and the Danish National Patient Register. In Denmark, every citizen is assigned a unique civil registration number. This number enables accurate linkage of information across different national registers. Information about gestational age as recorded in the National Patient Register was used throughout the study since clinical classification of births was based on this estimate. It is a clinical evaluation of gestational age at time of birth or pregnancy termination and widely based on ultrasound measures. In accordance with WHO definitions [18], we defined stillbirth as intrauterine death after ≥22 GA (gestational age) weeks and further divided it into early stillbirths (≥22–28 GA weeks) and late stillbirths (≥28 GA weeks). Neonatal death was death within 28 days of life. Children referred to as living were liveborn ≥22 GA weeks and lived beyond a minimum of 28 days.

### 2.2. Cause of Death Classification

For all late stillbirths and neonatal deaths in the cohort, medical records on both mothers and children, including autopsy results, were collected in 2003–2008 for the purpose of determining the primary cause of death. At the time, early stillbirths were classified as miscarriages, and thus only little information was available.

The primary cause of death was classified according to a classification system developed by Andersen et al. [16]. This system is based on two British classification systems by Cole et al. and Hey et al. [14,15] and seeks to combine fetal, obstetric and neonatal factors into 9 specific cause-of-death categories (See Table 1 and Appendix A for more detail). All stillbirths were retrieved and independently classified by two of the authors (E.A.N. and B.H.B), for the first 2/3 in 2005, and the last 1/3 in 2013. In cases of doubt or disagreement, the records were further evaluated by E.A.N., B.H.B., and A-M.N.A. until an agreement was reached. Neonatal deaths were classified by S.W. and E.A.N. in 2013.

As we wanted to be able to study stillbirth and neonatal death related to placental causes, Andersen et al.’s category “feto-placental dysfunction” was modified a priori [16]. Thus, “umbilical cord complications” were placed in a separate category, whereas “intrauterine growth restriction” (IUGR), “infarction of the placenta without IUGR”, “abruptio placentae” and “placenta previa” were combined into a single category: “placental dysfunction”. Andersen et al. include only antepartum deaths in the category “feto-placental dysfunction”; however, we classified neonatal deaths in this category if IUGR, abruptio placentae or placenta previa was the possible cause of neonatal demise. IUGR was defined as birth weight more than two standard deviations below the mean for gestational age on the reference curve suggested by Marsal et al. [19].

### 2.3. Exposures and Covariates

The main exposure was prepregnancy BMI (weight in kilograms/height in meters squared), based on self-reported information on prepregnancy weight and height from the first pregnancy interview, which was carried out at approximately 17 GA weeks (interquartile range 14–20 weeks). BMI was categorized according to WHO’s definitions as: underweight (BMI < 18.5), normal weight (18.5 ≤ BMI < 25), overweight (25 ≤ BMI < 30), and obesity (BMI 30+) [20]. Early gestational weight gain (GWG) was reported as a rate/week in grams. It was calculated as the difference between self-reported weight at the first interview and prepregnancy weight for women who provided a first interview between 8 and 22 GA weeks and divided by the length of pregnancy in weeks at the first interview. It was categorized into “lost weight or gain < 50 g”, “gain 50–300 g”, and “gain 300 g+”.

Other covariates, based on information from the first interview, included parity, smoking, and social status (a combination of education and occupation [3]). Pregnancy complications associated with obesity, defined as preeclampsia, hypertension and diabetes, were identified in the National Patient Register by their ICD 10 codes (International Classification of Diseases). As we expected some underreporting of gestational diabetes, we added self-reported information on this disease from the pregnancy interviews. The categorization of these variables is displayed in Table 2.

Of 100,413 pregnancies initially included in the DNBC, we excluded all multiple pregnancies (*n* = 2080), all pregnancies ending before 22 GA weeks (*n* = 5147), 10 pregnancies with unknown outcome, and 6 pregnancies with unknown gestational age. This first left us with 93,177 pregnancies. We further excluded pregnancies with no first interview (*n* = 5899) or without information about BMI in the first interview (*n* = 1456) This led to a final study population of 85,822 pregnancies of 80,136 women.

### 2.4. Statistical Analyses

Maternal and obstetric characteristics according to BMI group, stillbirth and neonatal death were presented with numbers and frequencies.

We used restricted cubic splines to examine the association between the main exposures and “stillbirth or neonatal death” which were combined into a single endpoint [21]. For prepregnancy BMI, we applied three knots (18.5, 25, 30) and used 20 as the reference value. For early GWG, we applied 5 knots based on the z-score distribution (−120 g, 80 g, 160 g, 250 g, 480 g) and used 160 g as reference equal to a z-score of 0. We also generated cubic splines stratified on prepregnancy BMI (<BMI 25 and ≥BMI 25). The splines for early GWG were used to guide the categorical analyses.

Cox regression analyses were used to estimate hazard ratios for the association between pre-pregnancy BMI and overall risk of stillbirth or neonatal death. We examined the association both for BMI in categories with normal weight as reference, and for continuous BMI in women with BMI ≥ 20 where cubic splines indicated a linear relationship. Follow-up started at 22 GA weeks (154 days) or at the day of the first interview if this was carried out later. Follow-up ended at the time of stillbirth, neonatal death, late induced abortion, emigration, maternal death before delivery, or after 28 days of life. We used the fetuses-at-risk approach with gestational age as survival time [22]. Thus, gestational age in days was the underlying timescale where survival in days was added for liveborn children. Neonatal deaths were censored at gestational age at birth plus the number of days they survived. To improve comparability, we stratified by gestational week at the first interview. We used the same approach to analyze all cause-of-death categories if they could include both stillbirths and neonatal deaths. For the category “early stillbirth”, follow-up started at 22 GA weeks and ended at 28 GA weeks. For the category “unexplained intrauterine death” (UID), follow-up started at 28 GA weeks and ended at birth, and for “preterm birth”, follow-up started at birth and ended after 28 days of life. In adjusted analyses, we controlled for maternal age as a continuous variable, and parity, smoking and social status. For cause-of-death categories with less than 40 deaths, we only adjusted for age and parity.

We used the same analytical approach for early GWG as the main exposure, but further controlled all adjusted analyses for BMI as a continuous variable. Early GWG of 50–300 g/week served as reference. As restricted cubic splines indicated that the association between early GWG and stillbirth or neonatal death differed according to BMI, we repeated the analysis within women with <BMI 25 and women with ≥BMI 25 and examined for interaction between BMI and GWG. We did not have statistical power to carry out stratified analyses of cause-of-death categories.

All analyses were repeated using logistic regression models. We also repeated all analyses after excluding women with obesity-related diseases in pregnancy. A correction for within-cluster correlation (robust standard errors) was applied to the final analyses, since 5616 women contributed more than one pregnancy to the study. We used STATA 16.0 Special Edition (Stata Corporation, College Station, Texas) for all statistical analyses.

## 3. Results

### 3.1. Causes of Stillbirth or Neonatal Death

We observed 633 deaths including 135 early stillbirths between 22 and 28 GA weeks, 272 late stillbirths after 28 GA weeks, and 226 neonatal deaths. Of the neonatal deaths, 36% happened within the first day of life and 77% within the first week of life. In 54% of deaths, autopsies had been made, and 57% had histopathological examinations of the placenta (81% for stillbirths).

The distribution of cause-specific death for this specific study is presented in absolute numbers and percentages in Table 1, and for the whole cohort in Appendix A, which also includes a more detailed presentation of the classification system. The most frequent causes of death were congenital anomalies (23.9%), UID (15.9%) and placental dysfunction (18.9%) (Table 1). Nine deaths (1.8%) were not classified because it was not possible to retrieve any information from medical records; they were included in the category “other causes”. Stillbirths comprised higher proportions of placental dysfunction (29.0%) and umbilical cord complication (29.0%), whereas neonatal death had higher proportions of congenital malformations (40.3%) and infections (10.2%). UID was only possible for stillbirths, just as the category “preterm birth” could only include neonatal deaths. For the category “preterm birth”, the data material did not allow us to confidently specify the specific cause of death.

### 3.2. Description of the Study Population

Of the mothers, 8% were categorized with obesity, 19% with overweight, 68% with normal weight, and 5% with underweight. Compared to women of normal weight, women with overweight and obesity were slightly younger, more often multiparous, more likely to smoke during pregnancy, less likely to be of highest social status, and more likely to suffer from diabetes or hypertensive disorders in pregnancy (Table 2). Women with overweight or obesity had much lower weight gains in early pregnancy (before 22 GA weeks) than other women. Thus, more that 29% of women with overweight and 54% of women with obesity had gained < 50 g/week. This gain category included many women with actual weight loss (24% with obesity, 12% with overweight, 4% with normal weight, and 2% with underweight).

Risk of stillbirth increased across BMI groups from 3.1/1000 births in underweight women to 7.5/1000 births in women with obesity. Risk of neonatal death was lowest in normal-weight women (2.2/1000 births) and highest in women with obesity (3.7/1000 births). For early GWG, women in the lowest category had the highest risks of stillbirth and neonatal death (7.1/1000 births and 3.0/1000 births, respectively)

### 3.3. Prepregnancy Obesity and Risk of Stillbirth or Neonatal Death

Restricted cubic splines indicated a J-shaped pattern between prepregnancy BMI and absolute risk of stillbirth or neonatal death (Figure 1a). After adjustment, risks increased with increasing BMI compared to women with a BMI of 20, while women at lower BMIs also tended to have a higher risk, however with wide confidence intervals including one (Figure 1b). For every one-unit increase in BMI in women with BMI ≥ 20, the risk increased with 5%, and crude and adjusted estimates did not differ much (Table 3). Compared to women of normal weight, women with overweight and obesity had increased risks of stillbirth or neonatal death of 66% and 78%, respectively. This pattern was also present for most cause-of-death categories. Thus, pronounced increased risks with increasing BMI were seen for UID where women with obesity had 2.5 times the risk, for deaths due to placental dysfunction with twice the risk in women with overweight and nearly 2.5 times the risk with obesity, and for deaths due to umbilical cord complications with 2.5 times the risk with obesity. Both for deaths due to intrapartum events and deaths due to infections, risks in women with obesity were 3.5 times the risk observed in normal-weight women. With increasing BMI, we also saw an increased risk of early stillbirth and death due to congenital anomalies, but most pronounced in women with overweight who had excess risks of 60% and 89%.

### 3.4. Early Gestational Weight Gain and Stillbirth or Neonatal Death

We observed a U-shaped pattern between early GWG and the absolute risk of stillbirth or neonatal death with higher risks with lower gains and an indication of slightly higher risk with very high gain, however with wide confidence intervals including one at the extremes (Figure 2a). This pattern was stronger in women with BMI < 25, but in women with BMI ≥ 25, no consistent association with early GWG was observed, and the overall absolute risk was at a somewhat higher level (Figure 2c,d). The same pattern was present in splines presenting adjusted hazard ratios, indicating a somewhat weaker association after controlling for BMI (Figure 2b–f).

In the adjusted analysis of categorical early GWG, we observed that compared to women who gained 50–300 g/week, women with low gain (lost weight or gained <50 g) had an 26% increased risk of overall stillbirth or neonatal death, while women with high gain (gained 300 g or more) had the same risk (Table 4). When we stratified the analysis on BMI, women with BMI < 25 and low gain had an increased risk of 43% of stillbirth or neonatal death, while this was only 13% in women with overweight or obesity with wide confidence intervals (p for interaction 0.21). High gain was not associated with increased risk in any of the BMI groups. In the analysis of cause-of-death categories, most associations with early GWG were estimated with large imprecision, but we observed that both women in the low and the high gain category had about 70% increased risk of early stillbirth. Additionally, a twofold risk of UID was seen in women with low gain compared to women who gained 50–300 g/week.

### 3.5. Supplementary Analyses

We repeated all analyses in women without obesity-related diseases in pregnancy, but these exclusions had only a minor influence on the observed associations. Additionally, results derived from logistic regression models were essentially the same as those derived from Cox regression.

## 4. Discussion

### 4.1. Main Findings

In this large nationwide Danish cohort study, we found increasing maternal prepregnancy BMI to be strongly associated with impaired fetal and neonatal survival. Especially in women with obesity, overall risk of stillbirth or neonatal death was increased, as were risks of UID and deaths due to placental dysfunction, umbilical cord complications, intrapartum events and infections. In women with overweight, we observed increased risk of death from all causes and specifically from congenital anomalies, placental dysfunction, infections and early stillbirth. For early GWG, we observed more stillbirths or neonatal deaths in women who were underweight or normal weight and had lost weight or gained < 50 g per week.

### 4.2. Prepregnancy Obesity

#### 4.2.1. Comparison with Other Studies

The observed increased risk of stillbirth and neonatal death in women with overweight and obesity concurs with a range of studies of different combinations of these outcomes, nicely summarized in a meta-analysis from 2014 [1]. In line with others [23], our data suggested a slightly increased risk in women with low BMI when combining the two outcomes, which may be a result of underweight women having a small increased risk of neonatal death [5,24], while decreasing BMI seems to be protective of stillbirth across the whole range of BMI [1]. Due to a limited number of women with severe obesity, we were not able to study the risk across categories of obesity, but large register studies suggest increasing risks with increasing severity, both for stillbirth [2,25] and neonatal death [24,26].

We used a classification system that allowed shared causes of death for stillbirth and neonatal death [16]. The same classification system has previously been used to study the association with maternal obesity, but without including neonatal deaths [27]. A study from the USA classified 658 stillbirths according to causes and also observed increased risks of deaths due to placental disease, umbilical cord complications and congenital anomalies in women with obesity [4]. In a Swedish study of cause-specific infant death, increased risks due to maternal obesity were observed for congenital anomalies, birth asphyxia, and other neonatal morbidity, but not for infections [6]. This corresponds well to a large US study where maternal obesity was associated with neonatal death caused by pregnancy complications or disorders relating to short gestation or low birth weight [5]. In other studies of neonatal death, the increased risk in mothers with obesity was highest for the first days/week of life [23,28], supporting that a large part of the increased neonatal mortality in the offspring of women with obesity have their origin in pregnancy or during delivery which justifies studying causes as a fetus–infant continuum across the perinatal period.

#### 4.2.2. Interpretation

The increased risk of death caused by placental dysfunction in women with obesity is supported by histopathological findings, observing far more placental lesions, especially those due to maternal vascular malperfusion, in stillbirths of women with obesity [29]. It is well accepted that women with obesity face higher risk of placental dysfunction, which may be explained by a disturbed metabolic and inflammatory profile including dyslipidemia, insulin resistance, hyperglycemia, low-grade inflammation, endothelial dysfunction, and oxidative stress [30,31,32]. As we did, other studies excluded women with pregnancy-related diseases and did not observe any change in estimates [6,23], indicating that women with obesity and no sign of clinical disease seem to face the same risk for stillbirth and neonatal death. There is no plausible explanation for more deaths due to umbilical cord complications in women with obesity as we and others have observed [4]. Umbilical cord entanglement and knots are common, even in uncomplicated births, and whether it is a causal factor is difficult to determine after death. Thus, some of these deaths may well be UIDs instead. We also observed more deaths caused by infection in obese women, which is in line with the above metabolic and inflammatory disturbances that may leave women with obesity more susceptible to infections [33], and maternal obesity has been associated with increased risks of both group B Streptococcus colonization in pregnancy [34] and early-onset neonatal bacterial sepsis [35]. More intrapartum deaths may be explained by the increased risk of pregnancy and birth complications in women with obesity, with higher risk of both very preterm and post-term births [36,37,38], macrosomia [39], operative deliveries, birth injuries, and birth asphyxia [40,41]. More deaths being caused by congenital anomalies fits well with more severe anomalies in offspring of mothers with obesity such as congenital heart anomalies [42,43] and neural tube defects [44] and may be attributable to nutritional and metabolic aberrations including folate deficiency [45] and hyperglycemia in early pregnancy [46].

### 4.3. Early Gestational Weight Gain

#### 4.3.1. Comparison with Other Studies

We could not identify any studies of the association between early GWG and stillbirth or neonatal death. Several large American register studies have investigated the association between total GWG and these outcomes [9,10,11], using the BMI-specific categories for GWG, recommended by the Institute of Medicine (IOM) [47]. We did not use IOM’s categories where excess gain in obese women is a total GWG > 9 kilos, but our findings are in line with two American studies that examined z-scores of actual total GWG. An increased risk of stillbirth was observed with low gain for all BMI groups, while indications of higher risk with very high GWG were only statistically significant in overweight women [48]. For neonatal death, U-shaped associations were observed for all BMI categories except for obesity grade 3 where there was no association [24]. As we did, several of the studies reported a higher background risk of stillbirth and neonatal death in women with obesity [9,11,24], which may complicate comparisons of relative risk estimates across BMI groups.

#### 4.3.2. Interpretation

In animal studies, a high fat diet has been associated with increased risk of stillbirth [7,8]. We used high early gestational weight gain in women as a measure for large gain in fat reserves which was supported by studies reporting associations between high early GWG and hyperglycemia [49] and gestational diabetes [50]. Our findings did not support that such an association also exists in humans; however, gain in fat reserves may not be a good proxy for a high fat diet. Instead, we observed that no gain or weight loss in early pregnancy increased risk of stillbirth or neonatal death, especially in underweight and normal-weight women. Low GWG has been associated with a range of adverse pregnancy outcomes such as preterm birth and small-for-gestational-age (SGA) [51], and we have recently reported that normal-weight women with low GWG are at increased risk of cardiovascular disease in long-term follow-up [52]. It may be that poor ability to gain weight during pregnancy may be a marker of a cardiovascular risk profile in the mother just as with SGA and preterm delivery [53], and that this risk profile increases risk of stillbirth or neonatal death. Additionally, miscarriages, stillbirth and neonatal death are more frequent in women that later develop cardiovascular disease [54,55].

We observed a substantially higher risk of UID in women who lost weight or had very low gain. Only little research has specifically looked at risk factors for UID, but inherited thrombophilia increased in women with UID [56] and a recent genetic study of UID fetuses suggests that UID may be caused by cardiogenetic pathologies [57]. We cannot explain our finding, which needs to be replicated in future studies, but it may be related to the cardiac risk profile in women with UID. Early stillbirth was the only outcome where we observed an increased risk in women with high early GWG. Since early stillbirths by definition were placed at the beginning of follow-up and right after early GWG was measured, it may be a result of reverse causation. Thus, pathology leading to the early stillbirth may also have resulted in high GWG. It is, however, a finding that deserves attention since early stillbirth is impossible to examine in studies of total GWG.

### 4.4. Strengths and Limitations

This study was based on a large cohort with nationwide data collection of medical records to determine causes of stillbirth or neonatal death. However, there are potential criticisms that need to be addressed. Information about prepregnancy BMI was self-reported and is likely to be underreported, which may result in an underestimation of the true BMI [58]. Since all women were weighed at the GP in early pregnancy, we assumed that this would reduce underreporting of their prepregnancy weight. Additionally, our measure of early GWG was the difference between two weights reported at the first interview, which will most likely provide more precise estimates than differences between weights reported at different points in time. All information was collected at time of entry into the study and therefore, it was not biased by information on the outcome of pregnancy. Information about stillbirths and neonatal deaths came from national registers or medical records, and any classification was carried out without knowledge to the woman’s exposure status. Thus, information bias, if present, would most likely lead to an underestimation of the true effect of obesity and early GWG.

As approximately 60% of those invited were recruited, the DNBC may on average be healthier than the general pregnant population. However, the prevalence of prepregnancy obesity was very close to the national estimates in the general female population aged 25–44 during the recruitment period, where 9.1% were found to be obese and 20.6% to be overweight [59]. Furthermore, a study of the impact of the initial selection on the internal comparisons in the cohort did not observe any selection bias of the association between prepregnancy BMI and stillbirth [60].

The fetus-at-risk approach that we applied in our statistical analysis, using gestational age as survival time, has been subject to controversy across many years [61,62]. Our classification of causes of stillbirth and neonatal death was based on the assumption of an overlapping etiology where most neonatal deaths have their cause in utero [14,15], which is supported by the observation that 77% of our neonatal deaths happened in the first week of life. Most of the cause categories could include both types of death or were time-specific, such as early stillbirth, preterm birth or intrapartum events. Proponents of the fetus-at-risk approach argue that the model is valid for causal inference (in this case, obesity and weight gain) and that it respects the fetus–infant continuum [22]. We repeated all analyses using logistic regression analyses with almost similar results, suggesting that at least the observed associations were not biased by the chosen approach. Another issue of our survival analysis was that we started follow-up at 22 GA weeks, which may introduce survival bias if obesity accelerates fetal demise early in pregnancy. We do not believe that this is a serious problem as we have observed obesity to be a weaker risk factor for fetal death in early pregnancy than later in gestation [3].

## 5. Conclusions

In this study, we found maternal prepregnancy overweight and obesity to be associated with overall stillbirth or neonatal death, and death due to specific causes such as placental dysfunction, umbilical cord complication, intrapartum events, and infection was more common in women with obesity. Although several biomarkers have been suggested to identify obese women at higher risk of pregnancy complications including stillbirth and neonatal death, none are considered clinically useful yet [63,64]. Ultrasound scans are today used for the screening of high-risk women, alone or in combination with biomarkers, but its specific application in stillbirth, especially in a broader population such as women with overweight or obesity, still needs to be proven in large clinical studies, which in itself is a great challenge due to the rarity of the outcome [64]. For now, the most obvious preventive measure is to enter pregnancy with a healthy weight, but how to accomplish this in childbearing women with obesity who want to have a baby is still a large and unsolved clinical challenge. Additionally, underweight or normal-weight women who lost weight or had very low gain in early pregnancy were at increased risk of stillbirth or neonatal death, and these women may deserve more attention in clinical care.

## Figures and Tables

**Figure 1 nutrients-13-01676-f001:**
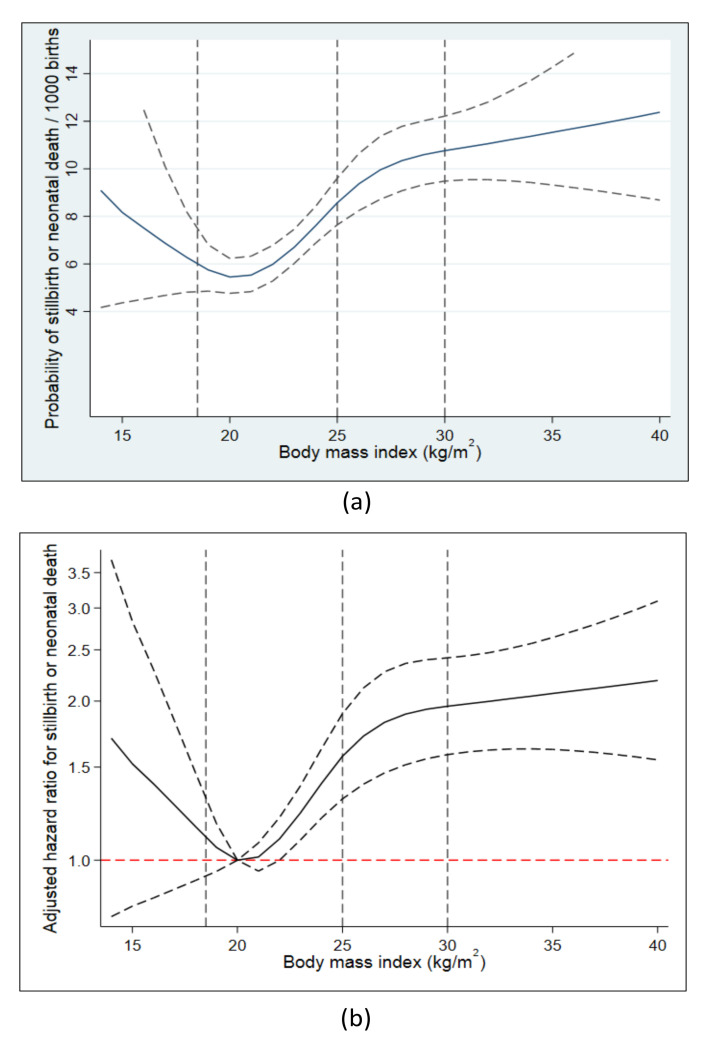
Absolute risks (**a**) and adjusted hazard ratios (**b**) for stillbirth or neonatal death according to prepregnancy BMI.

**Figure 2 nutrients-13-01676-f002:**
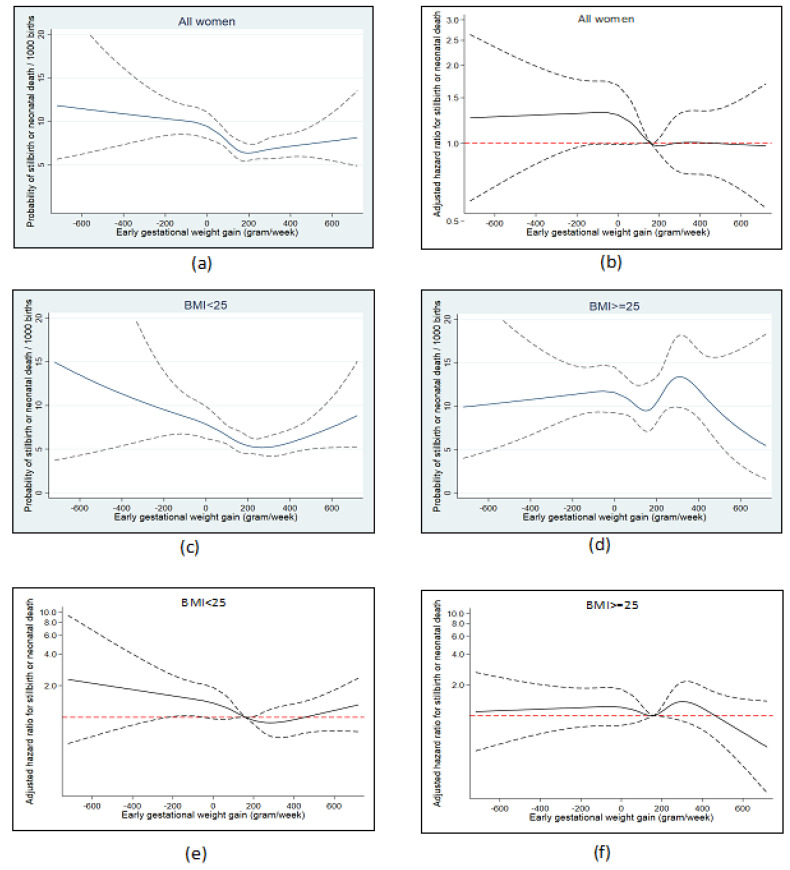
Absolute risks and adjusted hazard ratios for stillbirth or neonatal death according to early gestational weight gain in all women (**a**,**b**) and in women with BMI < 25 (**c**,**e**) and BMI ≥ 25 (**d**,**f**).

**Table 1 nutrients-13-01676-t001:** Distribution of cause-specific stillbirth and neonatal death in 272 late stillbirths ≧ 28 GA weeks and 226 neonatal deaths ^a^.

	Stillbirth		Neonatal Death		All Deaths	
Cause of Death	*n*	%	*n*	%	*n*	%
Congenital anomalies	28	10.3	91	40.3	119	23.9
Unexplained intrauterine death (UID)	79	29.0			79	15.9
Placental dysfunction	79	29.0	15	6.6	94	18.9
Umbilical cord complications	36	13.2	2	0.9	38	7.6
Maternal disease	9	3.3	12	5.3	21	4.2
Intrapartum events	23	8.5	26	11.5	49	9.8
Preterm birth			32	14.2	32	6.4
Infections	10	3.7	23	10.2	33	6.6
Other causes ^b^	8	2.9	25	11.1	33	6.6
Total	272	100	226	100	498	100

^a^ Among 85,822 pregnancies ≧ 22 GA weeks in the Danish National Birth Cohort. The study also included 135 deaths categorized as “early stillbirth” between 22 and 28 GA weeks. They were not part of the cause classification. ^b^ Other causes contained other specific conditions (23) and unclassifiable (10).

**Table 2 nutrients-13-01676-t002:** Maternal and neonatal characteristics of 85,822 pregnancies ^a^ by body mass index.

	All Pregnancies (*n*)	Body Mass Index					
		<18.5	18.5–24.9	25–29.9	30+	Stillbirth ^b^	Neonatal Death ^c^
Population (*n*)	85,822	3869	58,183	16,699	7071		
Stillbirth ^b^ (risk/1000)	407	3.1	4.0	6.7	7.5	4.7	
Neonatal death ^c^ (risk/1000)	226	3.4	2.2	3.5	3.7		2.6
**Age at conception**							
<25	11,080	18.5	11.9	13.8	16.3	4.3	3.2
25–29	35,770	41.8	41.6	41.9	41.8	3.9	2.4
30–34	29,131	30.2	34.6	33.3	31.8	4.8	2.5
35+	9841	9.5	11.9	11.1	10.1	8.0	2.9
**Parity**							
Primiparous	40,238	46.0	48.0	44.3	44.2	5.2	3.1
Multiparous	45,584	54.0	52.0	55.7	55.8	4.3	2.3
**Smoking**							
Nonsmoker or cessation	72,914	74.8	86.0	84.4	83.5	4.4	2.6
Smoking at 1st interview	12,878	25.2	14.0	15.6	16.6	6.5	2.9
**Social status**							
Highest	45,108	47.6	57.1	46.2	35.8	4.3	2.2
Middle	32,260	39.3	35.0	42.8	47.4	4.8	2.9
Lowest	8097	13.2	7.9	11.0	16.8	6.7	4.3
**Obesity-related disease**							
Pre-gestational diabetes	256	0.0	0.3	0.4	0.5	3.9	7.8
Gestational diabetes	1025	0.4	0.6	1.7	5.0	3.9	2.0
Preeclampsia	1905	0.9	1.7	3.0	5.4	7.3	6.8
Other hypertensive disorders	1359	0.6	1.1	2.2	4.4	3.7	5.2
**Early gestational weight gain** **^d^** **(gram/week)**							
Lost weight or gain < 50 g	13,903	7.2	14.6	29.3	53.7	7.1	3.0
50–300 g	41,598	64.4	66.0	54.7	36.5	4.1	2.8
300 g+	12,446	28.4	19.4	16.0	9.8	4.3	2.7

^a^ The actual study population with a first interview and information on body mass index. Numbers are column percentages except the first column, three first rows, and the last two columns. The number of missing values was 30 for smoking, 357 for socio-occupational status and 17,875 for early gestational weight gain. ^b^ Stillbirth was defined as fetal death after 22 completed weeks of gestation. ^c^ Neonatal death was defined as death of a liveborn infant within the first 28 days of life. ^d^ A weekly weight gain from prepregancy to time of first interview was estimated in 67,947 singleton pregnancies with a 1st interview between 8 and 22 GA weeks.

**Table 3 nutrients-13-01676-t003:** Risk of cause-specific stillbirth ^a^ or neonatal death ^a^ according to prepregnancy body mass index.

		BMI < 18.5 (3869)			BMI 18.5–24.9 (58,183)		BMI 25–29.9 (16,699)			BMI 30 + (7071)			Per 1 BMI-Unit Increase (Only BMI ≥20)	
*n* = 85,822 Cause (Number of Deaths)		Risk /1000	HR ^b^	95% CI	Risk/1000	HR Reference	Risk/ 1000	HR ^b^	95% CI	Risk/ 1000	HR ^b^	95% CI	HR ^b^	95% CI
All causes (633)	Crude	6.46	1.06	(0.70–1.58)	6.15	1.0 (ref)	10.24	1.66	(1.38–1.99)	11.17	1.82	(1.42–2.32)	1.05	(1.03–1.07)
	Adjusted ^c^		1.06	(0.70–1.60)		1.0 (ref)		1.66	(1.38–2.00)		1.78	(1.39–2.27)	1.05	(1.03–1.06)
Early stillbirths (135)	Crude	1.55	1.14	(0.50–2.62)	1.36	1.0 (ref)	2.22	1.63	(1.10–2.41)	1.84	1.34	(0.75–2.42)	1.03	(1.00–1.07
	Adjusted ^c^		1.08	(0.47–2.52)		1.0 (ref)		1.60	(1.08–2.38)		1.27	(0.70–2.29)	1.03	(0.99–1.07)
Congenital	Crude	2.07	1.91	(0.91–3.98)	1.10	1.0 (ref)	2.16	1.96	(1.30–2.94)	1.56	1.40	(0.74–2.65)	1.04	(1.01–1.07)
anomalies (119)	Adjusted ^c^		1.90	(0.90–4.02)		1.0 (ref)		1.89	(1.26–2.86)		1.34	(0.71–2.52)	1.03	(1.00–1.06)
Unexplained intra-	Crude	0.26	0.33	(0.05–2.38)	0.79	1.0 (ref)	1.14	1.43	(0.84–2.44)	1.84	2.31	(1.25–4.27)	1.05	(1.01–1.10)
uterine death (UID) (79)	Adjusted^c^		0.35	(0.05–2.56)		1.0 (ref)		1.50	(0.88–2.56)		2.42	(1.32–4.44)	1.06	(1.01–1.10)
Placental	Crude	1.03	1.25	(0.45–3.48)	0.82	1.0 (ref)	1.68	2.03	(1.27–3.24)	1.98	2.40	(1.33–4.34)	1.09	(1.05–1.12)
dysfunction (94)	Adjusted ^c^		1.21	(0.43–3.43)		1.0 (ref)		2.06	(1.27–3.34)		2.39	(1.30–4.39)	1.09	(1.05–1.12)
Umbilical cord	Crude	0.00	NA	NA	0.36	1.0 (ref)	0.66	1.84	(0.89–3.81)	0.85	2.40	(0.97–5.94)	1.08	(1.01–1.15)
complications (38)	Adjusted ^c^		NA	NA		1.0 (ref)		1.91	(0.93–3.93)		2.55	(1.02–6.37)	1.08	(1.02–1.15)
Maternal	Crude	0.52	2.52	(0.56–11.37)	0.21	1.0 (ref)	0.36	1.72	(0.64–4.60)	0.14	0.68	(0.09–5.22)	0.99	(0.91–1.08)
disease (21)	Adjusted^c^		2.81	(0.64–12.28)		1.0 (ref)		1.87	(0.72–4.83)		0.75	(0.10–5.70)	1.00	(0.93–1.08)
Intrapartum	Crude	0.00	NA	NA	0.50	1.0 (ref)	0.60	1.20	(0.58–2.46)	1.41	2.89	(1.41–5.93)	1.07	(1.02–1.13)
events (49)	Adjusted ^c^		NA	NA		1.0 (ref)		1.33	(0.64–2.75)		3.37	(1.59–7.12)	1.08	(1.03–1.14)
Preterm birth (32)	Crude	0.00	NA	NA	0.38	1.0 (ref)	0.36	0.94	(0.38–2.33)	0.57	1.54	(0.53–4.48)	1.03	(0.95–1.11)
	Adjusted ^c^		NA	NA		1.0 (ref)		0.94	(0.38–2.29)		1.52	(0.53–4.36)	1.03	(0.95–1.11)
Infections (33)	Crude	0.52	2.08	(0.48–9.04)	0.26	1.0 (ref)	0.60	2.31	(1.04–5.16)	0.85	3.30	(1.28–8.50)	1.08	(1.03–1.13)
	Adjusted ^c^		2.29	(0.54–9.78)		1.0 (ref)		2.43	(1.08–5.47)		3.56	(1.38–9.13)	1.08	(1.03–1.13)
Other causes ^d^ (33)	Crude	0.52	1.37	(0.32–5.70)	0.38	1.0 (ref)	0.48	1.26	(0.56–2.85)	0.14	0.38	(0.05–2.83)	0.96	(0.87–1.06)
	Adjusted ^c^		1.29	(0.30–5.50)		1.0 (ref)		1.24	(0.55–2.81)		0.37	(0.05–2.70)	0.96	(0.86–1.06)

^a^ Early stillbirths were fetal deaths 22–28 GA weeks. Other causes may include both stillbirths 28 + GA weeks and neonatal deaths (death in the first 28 days of life) except for UID (only stillbirths 28 + GA weeks) and preterm birth (only neonatal deaths). ^b^ Specific cause of death compared to all other births. ^c^ Causes >40 deaths were adjusted for age, parity, and social status. Causes <40 deaths were adjusted for BMI, age, and parity. ^d^ Other causes contained other specific conditions (23) and unclassifiable (10). BMI = body mass index; HR = hazard ratio; CI = confidence interval; NA = not applicable.

**Table 4 nutrients-13-01676-t004:** Risk of cause-specific stillbirth ^a^ or neonatal death ^a^ according to early gestational weight gain ^b^.

		<50 g/Week (13,899)			50–300 g/Week (41,578)		300 g+/Week (12,430)		
*n* = 67,947		Risk/1000	HR^c^	95% CI	Risk/1000	HR ^c^ Reference	Risk/1000	HR^c^	95% CI
Cause (number of deaths)									
All causes (513)	Crude	10.1	1.45	(1.18–1.78)	6.9	1.0 (ref)	6.9	1.00	(0.79–1.28)
*n* = 67,947	Adjusted ^d^		1.26	(1.01–1.58)		1.0 (ref)		0.95	(0.74–1.21)
BMI < 25 (303)	Crude	8.7	1.44	(1.07–1.94)	5.9	1.0 (ref)	5.5	0.94	(0.69–1.28)
n = 48,894	Adjusted ^d^		1.43	(1.05–1.93)		1.0 (ref)		0.90	(0.66–1.22)
BMI ≥25 (210)	Crude	11.6	1.10	(0.82–1.49)	10.3	1.0 (ref)	11.9	1.15	(0.77–1.72)
n = 19,053	Adjusted ^d^		1.13	(0.83–1.55)		1.0 (ref)		1.10	(0.73–1.64)
Cause-specific death, n = 67,947									*p for interaction = 0.21*
Early stillbirths (112)	Crude	2.2	1.82	(1.14–2.91)	1.3	1.0 (ref)	2.3	1.87	(1.19–2.95)
	Adjusted ^d^		1.71	(1.03–2.85)		1.0 (ref)		1.72	(1.09–2.72)
Congenital	Crude	1.6	1.10	(0.67–1.78)	1.4	1.0 (ref)	0.9	0.64	(0.34–1.23)
anomalies (91)	Adjusted ^d^		0.98	(0.57–1.69)		1.0 (ref)		0.66	(0.35–1.27)
Unexplained intra-	Crude	1.8	2.26	(1.31–3.90)	0.7	1.0 (ref)	0.3	0.42	(0.15–1.19)
uterine death (UID) (59)	Adjusted ^d^		2.07	(1.17–3.65)		1.0 (ref)		0.39	(0.14–1.11)
Placental	Crude	1.5	1.33	(0.79–2.23)	1.1	1.0 (ref)	0.8	0.72	(0.36–1.44)
dysfunction (77)	Adjusted ^d^		0.96	(0.52–1.76)		1.0 (ref)		0.64	(0.32–1.30)
Umbilical cord	Crude	0.6	1.42	(0.57–3.55)	0.3	1.0 (ref)	0.5	0.91	(0.30–2.73)
complications (28)	Adjusted ^d^		1.09	(0.40–2.94)		1.0 (ref)		0.88	(0.29–2.64)
Maternal	Crude	0.3	1.40	(0.45–4.35)	0.2	1.0 (ref)	0.4	2.03	(0.66–6.26)
disease (17)	Adjusted ^d^		1.39	(0.46–4.21)		1.0 (ref)		2.02	(0.66–6.17)
Intrapartum	Crude	0.7	1.05	(0.49–2.26)	0.6	1.0 (ref)	0.6	0.98	(0.44–2.19)
events (45)	Adjusted ^d^		0.75	(0.34–1.66)		1.0 (ref)		0.99	(0.44–2.21)
Preterm birth (28)	Crude	0.4	1.15	(0.44–3.00)	0.5	1.0 (ref)	0.2	0.52	(0.15–1.79)
	Adjusted ^d^		1.13	(0.42–3.03)		1.0 (ref)		0.52	(0.15–1.80)
Infections (29)	Crude	0.7	1.82	(0.82–4.03)	0.4	1.0 (ref)	0.2	0.62	(0.18–2.17)
	Adjusted ^d^		1.60	(0.63–4.07)		1.0 (ref)		0.61	(0.17–2.15)
Other causes ^e^ (27)	Crude	0.3	0.88	(0.27–2.75)	0.3	1.0 (ref)	0.7	2.22	(0.96–5.12)
	Adjusted ^d^		0.95	(0.31–2.91)		1.0 (ref)		2.22	(0.96–5.13)

^a^ Early stillbirths were fetal deaths 22−28 GA weeks. Other causes may include both stillbirths 28 + GA weeks and neonatal deaths (death in the first 28 days of life) except for UID (only stillbirths 28+ GA weeks) and preterm birth (only neonatal deaths). ^b^ GWG defined as the difference between prepregnancy weight and weight at first interview in women with interview between 8−22 GA weeks. Presented as an average weekly rate. ^c^ Specific cause of death compared to all other births. ^d^ Causes >40 deaths were adjusted for BMI, age, parity, and social status. Causes <40 deaths were adjusted for BMI, age, and parity. ^e^ Other causes contained other specific conditions (23) and unclassifiable (10). GWG = gestational weight gain; BMI = body mass index; HR = hazard ratio; CI = confidence interval; NA= not applicable.

## Data Availability

The data that the findings of this study are based on came from the Danish National Birth Cohort, and restrictions apply to these data, which were used under license for the current study and are not publicly available. The Danish National Birth Cohort welcomes requests for data which must include a short protocol with a specific research question and a plan for analysis. More information can be found at the website of the cohort [65].

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
