# Peer review of "Cause-Specific Stillbirth and Neonatal Death According to Prepregnancy Obesity and Early Gestational Weight Gain: A Study in the Danish National Birth Cohort"

_nutrients, 2021, doi:10.3390/nu13051676_

Round 1
Reviewer 1 Report
This interesting paper.
Language
The language seems clear enough to me. I am not a native speaker, however.
Originality
The association between pre gestational obesity and poor perinatal outcome was already known, but data from a large cohort are welcome. Besides, the attempt at grouping cases according to the likely cause death – either prenatal or neonatal death- is relevant. Trying to analyse the respective effect of pre-gestation BMI and early gestational weight gain is of interest. For all these reasons, I believe the data presented to be original enough.
Methods
The methodology looks fine.
Please explain the time gap between the time when enrolment took place and when the manuscript was written.
Please define more precisely the gestational age limits. I suppose we have <28 and ≥ 28 (line 86)
Line 120: please state that GWG reads per week (if this is correct)
Cubic splines:
I am no experts in statistics, but I shall comment on the interpretation of the results in the result section
Results
I have a major question: how can we interpret Figure 1 b ?
I would tend to say that with BMI grossly uneder22, there is no statistically significant relationship between BMI and mortality. However, there is a significant positive relation when BMI is greater than 22 or so. Am I right?
Similarly, on figure 2 b, the confidence interval of the adjusted risk of death seems to always “include 1” in the all women group (in which there should be a majority of non-obese persons). Do we agree that it would mean there is no statistical significance?
Same question for figure 2e and 2f
If I were not mistaken, then the meaning (and may be the title) of the paper would be: “pre pregnancy BMI, not first trimester GWG is related to perinatal mortality”? The paper would-be nonetheless of great interest.
If I were wrong, then it should be clearly stated why so, for instance in the discussion section. Sorry I my misunderstanding resulted from my ignorance, but it is likely the “average reader of the Journal” might be equally puzzled.
This may be all about the limits of interpreting non-significant “trends”.
Discussion
Could be made shorter and simpler
Conclusion
Very valuable data, but not presented and interpreted as they are.
I suggest the paper be resubmitted after thoroughly edited.
Reviewer 2 Report
The following are comments about the article Cause-specific stillbirth and neonatal death according to prepregnancy obesity and early gestational weith gain: a study in the Danish National Birth Cohort.
This is a large prospective cohort study that gives light to the association between obesity, gestational weight gain and perinatal mortality.
1. Introduction: the main results and references are correctly summarized.
2. Materials and Methods: covariates of the logistic regression models should be included in the Statistical analyses section.
3. Results: How would the authors explain the low prevalence of gestational diabetes and hypertensive disorders in this population?
4. Discussion: A high risk of stillbirth and neonatal death was detected between women with low gestational weight gain. Could an explanatory hypothesis be made by the authors?
This prospective study shows relevant associations between maternal overweight and obesity and stillbirth or neonatal death. The study reveals that placental dysfunction and infections are contributing factors to this excess of perinatal mortality. These factors are related to maternal obesity. Congratulations to the authors for this paper that clarifies the causes of perinatal mortality in mothers with overweight or obesity.
